# Atypical Presentation of Rapidly Progressive Cutaneous Metastases of Clear Cell Renal Carcinoma: A Case Report

**DOI:** 10.3390/medicina60111797

**Published:** 2024-11-01

**Authors:** Carmen Andrada Iliescu, Cristina Beiu, Andreea Racoviță, Cristina-Mihaela Olaru, Irina Tudose, Andreea Vrancianu, Liliana Gabriela Popa

**Affiliations:** 1Clinic of Dermatology, Elias Emergency University Hospital, 011461 Bucharest, Romania; carmenandrada1997.iliescu@gmail.com (C.A.I.); andreea.stefania.racovita@gmail.com (A.R.); liliana.popa@umfcd.ro (L.G.P.); 2Department of Oncologic Dermatology, Elias Emergency University Hospital, Carol Davila University of Medicine and Pharmacy, 020021 Bucharest, Romania; 3Department of Oncology, Elias Emergency University Hospital, Carol Davila University of Medicine and Pharmacy, 020021 Bucharest, Romania; mihaela.olaru@drd.umfcd.ro; 4Pathology Department, ‘Elias’ University Emergency Hospital, 011461 Bucharest, Romania; irina_tds@yahoo.com (I.T.); lynks_2111@yahoo.com (A.V.)

**Keywords:** cutaneous metastases, clear cell renal carcinoma, skin metastasis mimickers, oncologic dermatology, dermoscopy

## Abstract

Cutaneous metastases from clear cell renal carcinoma (ccRC) are uncommon and often indicate a poor prognosis. These metastases typically occur on the scalp, face, and trunk, and they can be difficult to diagnose due to their resemblance to benign dermatological tumors. We report the case of a 56-year-old patient with a history of ccRC (TNM stage 4) who was referred to our dermatology department with two rapidly enlarging, painful lesions on the left jawline and scalp, which had developed one month and one week earlier, respectively. On examination, the lesions appeared as well-defined, round to oval plaques with a central ulceration and a peripheral red rim, suggestive of an inflammatory appearance. Dermoscopic examination revealed a structureless pink to orange pattern, atypical central vessels, and irregular linear vessels in a corona-like arrangement. Despite the patient’s stable oncological treatment for six months, pain management had recently included paracetamol, tramadol, and NSAIDs. The primary presumptive diagnosis was of cutaneous metastasis, considering the patient’s history of metastatic ccRC. However, given the recent initiation of new pharmacological agents, the rapid progression of the cutaneous lesions, and their clinical presentation, alternative differential diagnoses were considered, including drug-induced reactions such as erythema multiforme or fixed drug eruption. A biopsy of the facial lesion revealed immunohistochemical positivity for CD10, CAIX, and PAX8, confirming the diagnosis of metastatic ccRC with sarcomatoid differentiation. Unfortunately, despite continued targeted therapies and palliative care, the patient’s condition deteriorated rapidly, leading to death two months later. This case highlights the potential for extremely rapidly evolving cutaneous metastases from ccRC and their capacity to occasionally mimic atypical drug eruptions. Additionally, it reaffirms the poor prognosis of such metastases, as evidenced by the patient’s death within two months.

## 1. Introduction

Renal cell carcinoma (RCC) accounts for approximately 3% of all adult malignancies, with the clear cell subtype being the most common, representing over 75% of cases [1,2]. While most patients are diagnosed with localized disease, up to 16% present with metastases at diagnosis [3]. The most common sites are the lungs, bones, and liver [4,5]. Skin metastases from RCC are exceedingly rare and usually manifest late in the disease [3]. Typically located on the scalp, face, and trunk, these lesions are often challenging to identify due to their tendency to mimic benign nodular dermatological tumors, such as angiomas, adnexal tumors, or pyogenic granulomas [6,7].

We present a unique case of rapidly progressive cutaneous metastases from ccRCC that mimicked an atypical drug eruption due to both clinical context—specifically, the recent initiation of new pain management therapies—and presentation, which strongly mimicked inflammatory plaques with central ulceration and a targetoid configuration. To our knowledge, this clinical presentation has not been previously reported in the literature.

This paper introduces a novel potential manifestation of skin metastasis from ccRCC and offers a comprehensive review of the existing literature on ccRCC cutaneous metastases.

## 2. Case Report

A 56-year-old male was referred to our dermatology department in July 2024 with two rapidly growing lesions: one located on the face, with a 4-week history, and a second on the scalp, with a 1-week history. Six years prior, the patient had been diagnosed with ccRC of the right kidney, stage pT3a pN0, Fuhrman grade 2, which had been treated by radical nephrectomy and pericave lymph node dissection. After three years of remission, the patient developed cerebral and lung metastases and received radiotherapy, followed by combination immunotherapy with Avelumab and Axitinib for six months, and subsequently cabozantinib. At the time of presentation, the patient’s treatment regimen included paracetamol, tramadol, and nonsteroidal anti-inflammatory drugs (NSAIDs), primarily for pain management.

On physical examination, two well-demarcated, round to oval targetoid plaques with a peripheral erythematous rim and a slightly ulcerated center, measuring 3 cm and 5 cm in diameter, respectively, were identified on the left mandibular region and scalp (Figure 1a,b). Both lesions elicited marked tenderness upon palpation, causing the patient significant discomfort.

The patient reported that the lesions initially presented as small erythematous plaques, which rapidly expanded through an active erythematous border, with the central area developing erosions that subsequently evolved into superficial ulcerations. This progression occurred over an approximate one-week period for each lesion.

Dermoscopy revealed a pink to orange, structureless pattern with a polymorphous vascular pattern in the central ulcerated area and linear-irregular vessels dispersed in a corona-like fashion in the periphery (Figure 2). 

Given the patient’s history of ccRC, the possibility of cutaneous metastases was considered as the primary differential diagnosis. However, the recent introduction of new medications, coupled with the rapid progression and the clinical and dermatoscopic characteristics of the lesions, also raised a strong suspicion of a drug-induced eruption, with erythema multiforme and fixed drug eruption as prominent considerations.

A punch biopsy from the edge of the facial lesion was performed to determine the diagnosis. While awaiting results, the recently introduced suspect medication was promptly discontinued, and the patient was advised to also avoid chemically related drugs. Topical management of the lesions included gentle cleansing with diluted chlorhexidine and twice-daily application of a medium-potency corticosteroid; however, this treatment provided no clinical benefit

The results of the biopsy were received 10 days post procedure. Histological analysis revealed an ill-defined tumoral proliferation involving both the dermis and hypodermis, predominantly composed of spindle cells and focally epithelioid cells (Figure 3a,b). The cells displayed large and hyperchromatic nuclei, irregular chromatin, and pale eosinophilic cytoplasm with poorly defined intercellular borders. The surrounding stroma, although sparse, was highly vascularised with mild lymphocytic inflammatory infiltrate. Mitotic figures were frequent, with some exhibiting atypical features (Figure 3c). No tumor necrosis was observed.

Immunohistochemical studies demonstrated diffuse expression of CD10, CAIX (Figure 4), and PAX8 (Figure 5) in the atypical cells. The tumor was consistently negative for p63 and S-100, further suggesting a potential sarcomatoid differentiation. The proliferation index within the lesion, as indicated by Ki-67, was estimated to be around 35% (Figure 6). These histopathological findings led to the diagnosis of cutaneous metastasis with sarcomatoid differentiation originating from the patient’s clear cell renal cell carcinoma. 

The patient was referred back to the oncology department, where the Multidisciplinary Tumor Board recommended continuation of cabozantinib therapy, initiation of immunotherapy with nivolumab, and palliative radiation therapy targeting the two cutaneous metastases. Despite these interventions, the disease continued to progress, and the patient ultimately passed away from cardiorespiratory arrest two months later.

## 3. Discussion

The incidence of skin metastases arising from RCC varies between 2.8% and 6.8% [8]. Most patients have a previously established diagnosis of RCC; however, a smaller subset may present with secondary skin involvement as the initial manifestation of an otherwise asymptomatic primary malignancy [9,10].

Cutaneous metastases from RCC are typically solitary and most commonly occur in the head, neck, and trunk regions, with less frequent involvement of the palms, soles, or nephrectomy scars [11]. They are more prevalent in males and generally develop within five months to several years after the initial diagnosis [6]. While our case shares many of these characteristics, it is notable for the presence of multiple metastatic lesions, making it an unusual presentation.

The precise mechanisms underlying cutaneous dissemination of RCC remain poorly understood, with only a limited number of studies addressing this issue [3,7,12,13]. Proposed pathways include hematogenous spread, lymphatic dissemination through the thoracic duct, direct invasion, and surgical implantation, with hematogenous dissemination considered the primary route [13]. The highly vascular nature of RCC facilitates tumor cell spread via the renal vein, leading to metastasis in multiple organs. For instance, pulmonary metastases occur as tumor cells travel from the renal vein to the inferior vena cava, and subsequently to the right atrium and lungs. This pulmonary infiltration route can be bypassed via arteriovenous and systemic shunts, allowing tumor spread to the head and neck region [12]. In our case, the presence of metastatic lesions on the scalp and mandibular region may suggest lymphohematogenous spread. Additionally, an alternative pathway involving the valveless vertebral veins (Batson’s plexus) could facilitate tumor cell migration from the renal vein to the emissary veins, ultimately reaching the scalp and skin [14]. Skin metastases of RCC exhibit variable clinical features, usually appearing as rapidly growing, nodular lesions, ranging in color from red to purple, accompanied sometimes by pain, bleeding, or ulceration [13,15,16]. These lesions often mimic benign tumors, such as angioma, pyogenic granuloma, and adnexal tumors, as well as malignant conditions such as basal cell carcinoma, Kaposi sarcoma, and cutaneous lymphoma [6,9,17]. In our case, the cutaneous metastasis exhibited an atypical presentation as inflammatory targetoid plaques. The lesions emerged following the recent introduction of new medications and progressed rapidly within one week, raising the possibility of a drug-induced eruption, such as erythema multiforme or fixed drug eruption [18,19].

To our knowledge, this is the first documented case of RCC cutaneous metastasis closely resembling a drug eruption. This case highlights the importance of maintaining a high index of suspicion and clinical vigilance, as the variable presentation of such metastases can significantly delay both diagnosis and appropriate treatment. Additionally, the suspected medications were promptly discontinued, and chemically related drugs were avoided to prevent potential cross-reactivity [20], a precaution that ultimately limited the patient’s options for pain management without benefit. 

Dermoscopy can offer valuable diagnostic clues and aid in raising clinical suspicion for RCC skin metastases, particularly when lesions present with ambiguous or atypical features. Recent studies have highlighted the importance of identifying a white, structureless pattern accompanied by linear, serpentine vessels as indicative of skin metastasis, especially in patients with a known history of neoplasia, as seen in our case [21]. Additionally, the high prevalence of vascular patterns in skin metastases may reflect underlying angiogenesis, a critical process in cancer progression [22]. In our case, a polymorphous vascular pattern (dotted, arborizing, and tortuous vessels) on dermoscopic examination was the key finding that raised suspicion for an alternative diagnosis beyond a drug eruption. Such findings are rarely seen in drug reactions, where vascular patterns are generally limited to short, linear vessels [23]. This highlights the diagnostic value of dermoscopy, although histopathological examination remains essential for definitive diagnosis.

Histologically, cutaneous metastases of RCC typically present as tumoral proliferations within the dermis and subcutis, characterized in up to 80% of cases by clear cells with abundant pale cytoplasm, large hyperchromatic nuclei, and conspicuous nucleoli [24,25]. In cases like ours, where spindle or epithelioid cells are present, the tumor is classified as the less common and more aggressive mixed clear cell and sarcomatoid subtype [24]. In addition to cellular morphology, immunohistochemical markers further refine the diagnosis. PAX2 and PAX8 are transcription factors with a high sensitivity for renal tissue, while CD10 is a membrane-bound protein frequently expressed in renal neoplasms [26,27]. Additionally, high tumoral expression of CAIX, a transmembrane protein present in most clear cell RCCs, has been associated with better prognosis and an improved likelihood of response to interleukin-2-based immunotherapy [28]. 

Given the limited therapeutic options and the aggressive nature of metastatic RCC, treatments are often palliative. For solitary cutaneous metastases, complete surgical excision is the preferred approach, especially in cases of rapid lesion growth, bleeding, or local invasion [10]. However, when surgical removal is not feasible, hypofractionated radiotherapy has proven effective in providing rapid palliation of skin lesions [29]. Recently, new systemic therapies, including immune checkpoint inhibitors such as nivolumab and ipilimumab, have emerged as promising treatments for advanced RCC [30]. Although these agents have improved overall survival, their effectiveness in managing cutaneous metastases remains uncertain [31]. In the case of our patient, he was not a candidate for surgery, and received palliative radiotherapy for the two cutaneous metastases, alongside newly initiated systemic immunotherapy with nivolumab.

Despite these interventions, the patient passed away two months later. The presence of skin metastases from RCC is well established as a poor prognostic indicator, with a reported mean survival time of 10.9 months following diagnosis [24]. In our case, the patient’s death occurred significantly earlier, further highlighting the poor prognosis. Given the rarity of cutaneous metastasis from ccRC, further data are needed to explore potential correlations between the clinical characteristics, progression of these lesions, and overall prognosis.

## 4. Conclusions

In conclusion, we present a rare case of cutaneous metastases from ccRC, highlighting the complexity in the presentation, progression, and prognosis of this form of skin metastasis. To our knowledge, this is the first documented case of ccRC cutaneous metastasis manifesting as rapidly progressive plaques with an inflammatory, targetoid appearance, which initially also raised the suspicion of an atypical drug eruption. Maintaining a high degree of clinical suspicion in such cases, coupled with thorough histopathological evaluation, is crucial to avoid diagnostic errors and ensure appropriate management. 

## Figures and Tables

**Figure 1 medicina-60-01797-f001:**
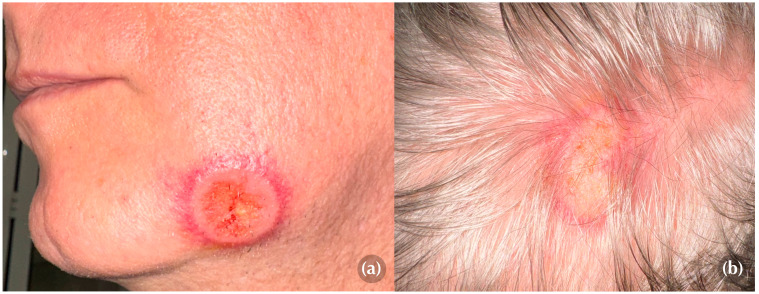
Clinical aspect of the lesions. (**a**) A well-defined, round plaque with a targetoid appearance, characterized by peripheral erythema, a pale edematous ring, and an ulcerated center located on the left mandibular region. (**b**) A similar but less well-defined lesion located on the scalp.

**Figure 2 medicina-60-01797-f002:**
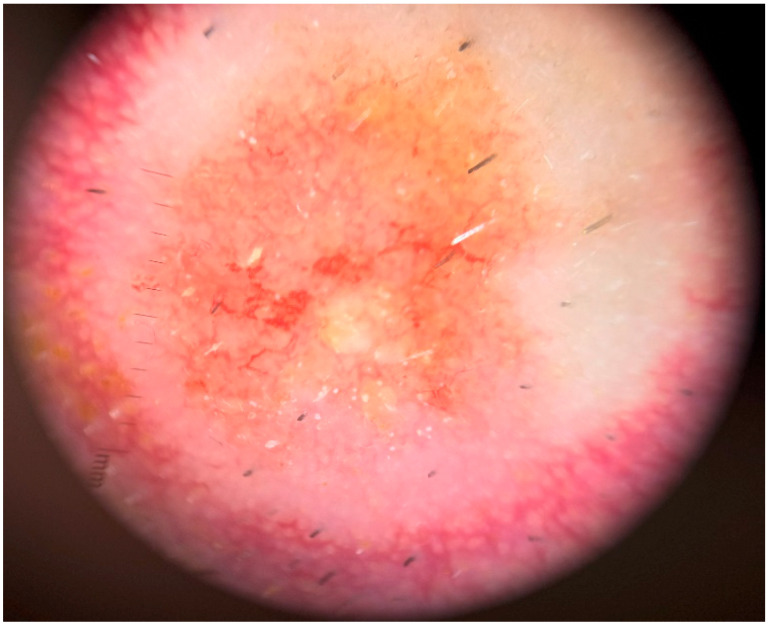
Dermoscopy of the lesion on the left mandibular region. Presence of a pink to orange structureless pattern with polymorphous vasculature (dotted, arborizing, and tortuous vessels) in the central ulcerated area along with linear-irregular vessels in a corona-like distribution at the periphery.

**Figure 3 medicina-60-01797-f003:**
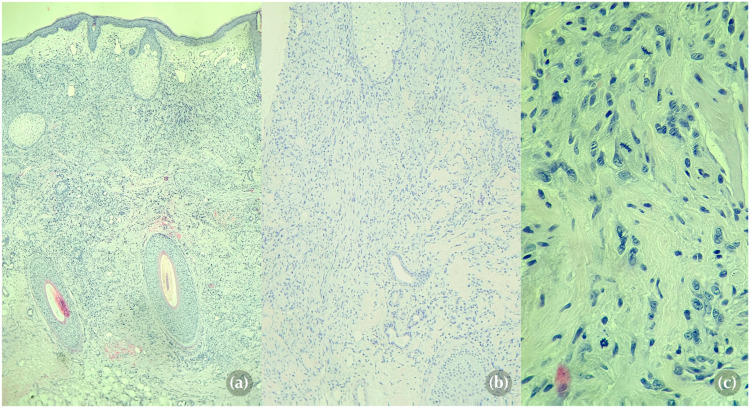
Histopathological findings from the punch biopsy sample. (**a**) Infiltrative spindle cell proliferation, developed at the dermo-hypodermic level (hematoxylin and eosin stain, original magnification ×40). (**b**) Spindle cells arranged in a vaguely fascicular pattern (hematoxylin and eosin stain, original magnification ×100). (**c**) Fusiform cells with numerous mitotic figures, some of which are atypical (hematoxylin and eosin stain, original magnification ×400).

**Figure 4 medicina-60-01797-f004:**
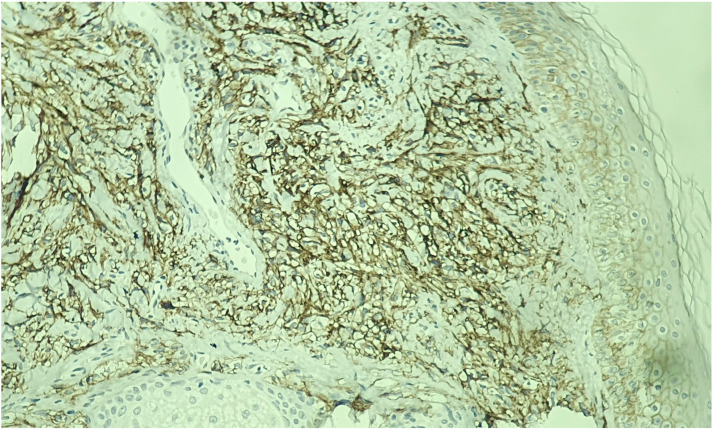
Immunohistochemistry showing a positive CAIX membranous pattern with a box-like appearance (×200 magnification).

**Figure 5 medicina-60-01797-f005:**
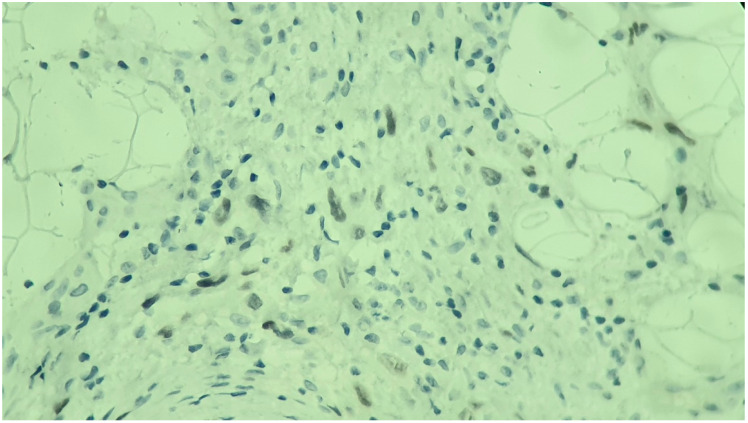
Immunohistochemistry showing nuclear positivity for PAX8 (×200 magnification).

**Figure 6 medicina-60-01797-f006:**
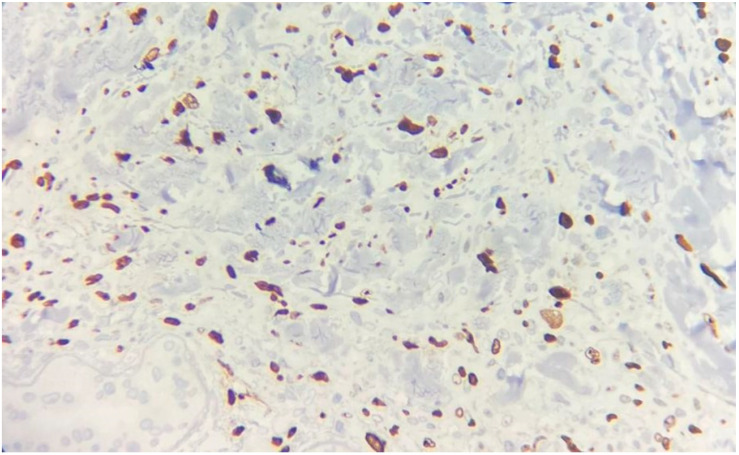
Proliferation index (Ki-67) was positive in approximately 35% of the nuclei of tumor cells.

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
