# Peer review of "Atypical Presentation of Rapidly Progressive Cutaneous Metastases of Clear Cell Renal Carcinoma: A Case Report"

_medicina, 2024, doi:10.3390/medicina60111797_

Round 1
Reviewer 1 Report
Comments and Suggestions for Authors
Dear Authors,
Thank you for your efforts in presenting this case report on cutaneous metastasis from clear cell renal carcinoma (ccRC).
While I appreciate the diagnostic challenges highlighted, I have a slightly different perspective regarding the primary differential. Given the patient's history of ccRC, metastatic cutaneous involvement could have been a strong primary differential from the outset. Nevertheless, as you correctly noted, the differential diagnosis of a drug-induced eruption was a valid consideration, and such differing viewpoints are common in clinical practice. However, since this differential was quickly ruled out, I find it somewhat unjustified to structure the entire manuscript, including the title, around this initial consideration—especially given the extensive documentation of diverse cutaneous presentations of ccRC in the literature, with an incidence of 3-7%.
Additionally, the conclusion, stating "This case highlights the aggressive progression and poor prognosis associated with cutaneous metastases from renal cell carcinoma," could risk overgeneralizing findings from a single case. It is essential to be cautious in making broad claims based on individual reports, as single cases may not fully represent the variability observed in clinical outcomes.
I recommend revising the title and content to provide a more balanced and comprehensive overview of cutaneous metastases in ccRC, covering their varied presentations, diagnostic challenges, treatment options, and prognostic implications. Additionally, please ensure the manuscript aligns with the CARE guidelines for case reports.
Best regards,
Comments on the Quality of English LanguageMinor proofreading is required.
Author Response
Dear Reviewer,
Thank you very much for your constructive feedback on our manuscript. We greatly appreciate the time and care you put into reviewing our work. Your insights helped us refine and strengthen the presentation of this case. We have made the following revisions based on your comments:
- The content has been revised to provide a more balanced and comprehensive overview of cutaneous metastases in ccRC. We have clarified that, given the patient’s history of ccRC, the possibility of cutaneous metastasis was considered as the primary differential diagnosis. However, the recent introduction of new medications, along with the rapid progression and clinical and dermatoscopic characteristics of the lesions, also raised a strong suspicion of a drug-induced eruption, specifically erythema multiforme and fixed drug eruption. The entire manuscript has been adjusted to avoid the impression that it was structured around this initial consideration.
- The title has also been revised to better reflect that this case represents an atypical, rapidly progressive form of cutaneous metastasis from ccRC.
- We have removed the statement in the conclusion that "This case highlights the aggressive progression and poor prognosis associated with cutaneous metastases from renal cell carcinoma." We agree that this could overgeneralize findings from a single case.
- We have ensured that the manuscript aligns with the CARE guidelines for case reports.
- We carefully proofread the manuscript and enlisted the assistance of a native English speaker to ensure clarity, accuracy, and fluency throughout the text.
Reviewer 2 Report
Comments and Suggestions for Authors
Dear authors, I find your topic interesting and not so uncommon in the clinic. it is right to highlight a similar phenomenon to create, with a broader series of cases, a flowchart capable of effectively resolving such a metastasis. The structure of the manuscript is well done, the arguments are well developed, the language is fluid and understandable and the results are clearly stated. Nothing to add.
Author Response
Dear Reviewer,
Thank you very much for your positive feedback. We are delighted that you found our topic interesting and appreciated the structure, language, and clarity of our manuscript. Your support reinforces the importance of documenting such cases to contribute to broader clinical insights and, ultimately, aid in developing a more effective diagnostic approach for similar metastases.
We sincerely appreciate your time and thoughtful comments, which motivate us to continue our work in this area.